# Core collection construction and genetic diversity analysis of tea plant (*Camellia sinensis* [L:] O. Kuntze) accessions in Huangshan city using SSR markers

Xu Ruan , Qiong Wu, Xiaoyu Jiao, Dandan Liu, Minghui Sun, Leigang Wang, Wenjie Wang *

Tea Research Institute, Anhui Academy of Agricultural Sciences, Hefei, Anhui, China,

* wwj00@126.com

## Abstract

Assessing genetic diversity and building a core collection is essential to advancing tea plant breeding. In this study, ten SSR markers exhibiting robust amplification and polymorphism were employed to genotype 292 tea accessions sourced from various regions in Huangshan city. The results revealed significant genetic variation, encompassing 180 alleles. Genetic structure was evaluated using neighbor-joining clustering, principal coordinate analysis, and Structure analyses, which categorized the tea accessions into two primary clusters. The genetic diversity within these clusters demonstrated high similarity, likely due to their close geographical proximity. A core collection was established utilizing Core Hunter software, resulting in the selection of 35% of the accessions to effectively represent the genetic diversity of the entire collection. This core collection comprises 102 tea accessions, preserving a high percentage of allele richness and genetic diversity. This research offers valuable insights into genomics research and the sustainable management of tea plant genetic resources in Huangshan city.

## Introduction

Tea (*Camellia sinensis* [L.] O. Kuntze) is a perennial woody plant and the world's most popular beverage crop, originating in southwest China with a rich history spanning over 3,000 years [1–3]. As a self-incompatible species, it exhibits significant genetic diversity due to prolonged natural selection [4]. The climate in Anhui province is particularly suitable for tea cultivation [5]. The region boasts an abundance of tea germplasm, with tea production primarily concentrated in the southern areas of the Yangtze River and the Dabie Mountains. Huangshan city, situated in Anhui province, is renowned for its long-standing tradition of producing famous teas such as Huangshan Maofeng, Taiping Houkui, and Keemun black tea, alongside a wealth of local tea landraces. However, the widespread adoption of clonal tea varieties has

**Data availability statement:** All relevant data are within the manuscript and its Supporting Information files.

**Funding:** The author(s) received no specific funding for this work.;

**Competing interests:** The authors have declared that no competing interests exist.

led to a decline in these landraces in Huangshan city in recent years. Furthermore, excessive harvesting of wild tea plants has contributed to a reduction in natural populations. To preserve resource diversity and support breeding research, it is essential to gather and conserve wild and landrace resources in Huangshan city. In response, the Tea Research Institute of Anhui Academy of Agricultural Sciences established the Anhui Landraces Germplasm Tea Repository in 2020. This initiative has successfully collected over 90,000 landrace seeds from 27 major tea-producing counties and districts, encompassing nearly the entire tea-producing area in Anhui province. The repository features separate rows designated for planting landraces from each area, with each row containing approximately 100 landraces individuals. Overall, the repository houses nearly 100,000 landraces individuals, including about 30,000 tea landraces individuals sourced from 300 villages within Huangshan city.

It is essential to construct a core collection that accurately represents genetic diversity while minimizing the number of accessions to effectively protect and utilize diverse landraces [6,7]. Currently, research on core collection has been conducted across various crops, including cotton [8], peanut [9], potato [10] and radish [11]. A core collection comprising 532 accessions was randomly selected from the China National Germplasm Tea Repository, based on cultivated region grouping and phenotypic data, with approximately 20% of the accessions included [12]. In 2004, 126 tea accessions were selected from a total of 615 tea plants in China as a core collection; genetic diversity and agronomic trait analyses indicated that this collection was well representative [13]. Recently, research on core germplasm collections has primarily relied on morphological and molecular markers [14]. The main molecular markers employed include Amplified fragment length polymorphism (AFLP), Random amplified polymorphic DNA (RAPD), Single nucleotide polymorphism (SNP), and Simple Sequence Repeats (SSR), among other technologies [15–17]. SSR is particularly favored due to its high polymorphism, substantial informational content, and co-dominant inheritance.

23 SSR markers were employed to analyze 788 tea accessions from around the world, resulting in the construction of a core collection comprising 192 accessions that encapsulated the variation present within the 788 accessions [18]. A targeted core collection was subsequently established using 33 SSR markers for 462 tea accessions [19], which included 100 accessions (21.6% of the total collection) originating from four regions: 73 from Korea, 22 from China, 4 from Japan, and 1 from India. Additionally, 171 tea accessions were characterized from Sri Lankan and Indian germplasms phenotypically and genotypically using 28 SSR markers, identifying 21 Sri Lankan and 18 Indian accessions as part of the core collections [20]. The population structure of 573 tea accessions in Anhui Province was analyzed using 60 SSR markers, leading to the development of a core collection of 115 accessions that effectively represented the genetic diversity of the original samples [21]. Notably, there remains a gap in research regarding the core collection of tea landraces in Anhui Province. In this study, 292 tea accessions were selected from Huangshan city, sourced from the Anhui Landraces Germplasm Tea Repository, comprising 287 landraces and 5 cultivars. To assess genetic diversity and population structure, ten

highly polymorphic SSR primers were utilized. The objective was to construct a core collection and provide insights for the conservation and utilization of tea accessions in Anhui province.

## Materials and methods

### Primer screening

Ten SSR markers were selected from the 74 pairs of tea plant SSR markers developed by Ma [22] (Table 1). Subsequently, 10 pairs of SSR primers were synthesized by Shanghai Sheng Gong Co., Ltd. (Shanghai, China) for further experiments.

### Plant materials

A total of 287 landraces were sampled from the Anhui Landraces Germplasm Tea Repository. They were collected from towns in She County (SX), Xiuning County (XN), Qimen County (QM), Yi County (YX), Huizhou District (HZ), and Huangshan District (HS). The number of towns was 13, 20, 16, 7, 5, and 8 respectively, and the number of selected villages of each town was 69, 48, 49, 34, 42, and 50 respectively. One landrace from each village was selected randomly. In addition, 5 cultivars, such as Wancha No. 4 (WC04), Wancha No. 5 (WC05), Anhui No. 1 (AH01), Anhui No. 3 (AH03), and Caoxi No. 1 (CX01) were included, bringing the total to 292 experimental materials (S1 Table).

### DNA extraction and PCR amplification

Young leaves of tea plants were collected and extracted using a Tiangen Express plant genomic DNA kit (Beijing, China). The quality of the extracted DNA was assessed via 1% agarose gel electrophoresis. The DNA was subsequently diluted to a concentration of 30 ng/µL and stored at −20°C for future experiments. For the PCR experiment, reagents were sourced from Jinsha Corporation (Beijing, China). The concentration of the PCR product was estimated using agarose gel electrophoresis; 1 µL of the product was diluted to 10 ng/µL with water. The mixture was prepared with an internal plate at a HiDi ratio of 1:130. The diluted 1 µL product was combined with the mixture at a ratio of 1:9, denatured at 95°C for 5 minutes, and then placed on the sample stand of an ABI3730XL sequencer (ABI, USA) for capillary electrophoresis detection.

Table 1.  Information of 10 SSR primers used in this study.

| Marker | Repeat motif | Forward primer sequence 5′→ 3′ | Reverse primer sequence 5′→ 3′ | Tm (°C) | Size range (bp) |
|---|---|---|---|---|---|
| TM045 | (TC)13 | ATCCTCATCCTCTTCGTCAT | AGATCCAGAACTTAGCAACA | 56 | 320–340 |
| TM058 | (TCA)6 | CATTATCCCTTTCCTTGTCCA | GGAGGGAGTAGGAGGTGGTCT | 58 | 250–270 |
| TM065 | (AG)11 | TTTGATGATCCATTAGTGTA | AGTACATCTATCCCAAAACA | 52 | 220–270 |
| TM069 | (CT)9 | CTGTCATGTTCTTGAGCTGT | GACCCATACTTTCATATTTG | 54 | 150–170 |
| TM072 | (TC)10 | AAGGCATTGTCCTCTTTTCC | CATTTGCTCACTTACCCCAT | 60 | 275–290 |
| TM075 | (CT)16 | ACCCCTGTTTTCGTCTTTAC | GTTGCTGTTGATTCGTCTGA | 58 | 135–155 |
| TM076 | (CT)9 | ATTCCCATTCACTTCAACA | CTAACTGAGCGAGCCCTCTT | 58 | 220–240 |
| TM092 | (CT)11 | TGCTAGGAGGCAAGGAGGCC | GACCCTTCCGAGGAACGAGA | 60 | 105–120 |
| TM099 | (TC)10 | CTGCCCGTGAAGTTAGTTTT | ATCGAATCAACCATTAGAAAGT | 56 | 230–255 |
| TM103 | (TATGTG)4 | GTCCCCATTGCTCTTAGTTT | ATCATTGACCACCACATCAT | 56 | 210–225 |

Tm: temperature of melting.

## Data analysis

GenAlEx v6.5 [23] was employed to calculate various genetic parameters for each microsatellite marker, including the number of alleles (Na), effective number of alleles (Ne), observed heterozygosity (Ho), expected heterozygosity (He), unbiased expected heterozygosity (Uhe), hybridization coefficient (F), and Shannon's Information index (I). The polymorphism information content (PIC) was determined using PowerMarker V3.25 software [24]. Additionally, F-statistical calculations (FIS, FIT, and FST) and principal coordinate analysis (PCoA) were conducted in GenAlEx v6.5, in conjunction with Microsoft Excel. Genetic distances were clustered using PowerMarker V3.25 [24] following the neighbor-joining clustering [25], and the resulting phylogenetic trees were enhanced and edited using the ITOL website [26].

Structure V2.3.4 [27] was employed for population structure analysis utilizing a Bayesian model-based hybrid analysis approach. The Markov chain Monte Carlo (MCMC) was configured with 100,000 iterations and a run length of 100,000, while the number of genetically homogeneous clusters (K value) was set to range from 2 to 20, with 10 repeated runs for each K value. The optimal K value was determined using the highest ΔK method as proposed by Evanno [28]. Subsequently, the optimal set of core germplasm collection was extracted using Core Hunter V3.0 [29], which employs a local search algorithm to maximize genetic diversity and allele richness. Ten sample components were tested (10%, 15%, 20%, 25%, 30%, 35%, 40%, 45%, 50%, and the initial component) with Core Hunter V3.0, based on the reported distribution of core germplasm components of woody plants [30–32]. T-tests were conducted between the core and initial germplasm groups using Microsoft Excel. The smallest core germplasm (P≤0.05) that was not significantly different from the 100% population group was selected to construct the optimal core collection [33].

## Results

### Genetic diversity of tea accessions

A total of 180 alleles were identified within 292 tea accessions using 10 SSR markers. Each tea plant was successfully genotyped with these alleles, demonstrating the strong discriminatory ability of the SSR markers (Table 2). The markers exhibited significant genetic variation, with the number of alleles per locus (Na) ranging from 15 to 22 (mean = 18.00), the effective number of alleles (Ne) from 4.00 to 9.71 (mean = 7.02), observed heterozygosity (Ho) from 0.14 to 0.61 (mean =

**Table 2. Genetic diversity parameters for the 292 accessions using SSR markers.**

| Marker | Na | Ne | I | Ho | He | uHe | F | PIC |
|---|---|---|---|---|---|---|---|---|
| TM045 | 22.000 | 8.833 | 2.509 | 0.360 | 0.887 | 0.888 | 0.595 | 0.877 |
| TM058 | 17.000 | 7.954 | 2.373 | 0.140 | 0.874 | 0.876 | 0.839 | 0.864 |
| TM065 | 20.000 | 6.018 | 2.114 | 0.199 | 0.834 | 0.835 | 0.762 | 0.815 |
| TM069 | 15.000 | 6.003 | 2.054 | 0.253 | 0.833 | 0.835 | 0.696 | 0.815 |
| TM072 | 19.000 | 7.954 | 2.335 | 0.315 | 0.874 | 0.876 | 0.640 | 0.862 |
| TM076 | 15.000 | 4.003 | 1.739 | 0.322 | 0.750 | 0.751 | 0.571 | 0.717 |
| TM077 | 21.000 | 9.716 | 2.558 | 0.312 | 0.897 | 0.899 | 0.653 | 0.889 |
| TM085 | 15.000 | 7.141 | 2.205 | 0.606 | 0.860 | 0.861 | 0.295 | 0.845 |
| TM092 | 16.000 | 7.980 | 2.322 | 0.562 | 0.875 | 0.876 | 0.358 | 0.863 |
| TM099 | 20.000 | 4.550 | 2.023 | 0.288 | 0.780 | 0.782 | 0.631 | 0.764 |
| **Mean** | 18.000 | 7.015 | 2.223 | 0.336 | 0.846 | 0.848 | 0.604 | 0.831 |
| **St.Dev** | (0.856) | (0.582) | (0.078) | (0.046) | (0.015) | (0.015) | (0.053) | (0.055) |

No.: number, N: number of individuals, Na: number of alleles, Ne: number of effective alleles, Ho: observed heterozygosity, He: expected heterozygosity, uHe: Unbiased Expected Heterozygosity = [2N/(2N−1)] × He, I: Shannon's information index, F: inbreeding coefficient = (He − Ho)/He = 1 − (Ho/He), PIC: polymorphic information content.

0.34), expected heterozygosity (He) from 0.75 to 0.90 (mean = 0.85), unbiased expected heterozygosity (uHe) from 0.75 to 0.90 (mean = 0.85), Shannon's diversity index (I) from 1.74 to 2.56 (mean = 2.22), and polymorphic information content (PIC) from 0.72 to 0.89 (mean = 0.83). The SSR marker TM045 exhibited the highest allele count (22), while TM069, TM076, and TM085 had the lowest (15). Observed heterozygosity was consistently lower than expected heterozygosity across all markers. The high level of polymorphism was evident, as indicated by a PIC value exceeding 0.7 for all markers. Notably, the TM077 marker proved particularly effective in capturing genetic diversity, with the highest observed PIC value of 0.89.

The genetic diversity index values displayed minimal variation in Huangshan city. The SX population exhibited the highest values for Na and I, while the HS population demonstrated the highest values for Ne and Ho, along with the lowest fixation index (F) (Table 3).

## Genetic structure of tea accessions

The genetic relationships among the test materials were examined by constructing a phylogenetic tree using the neighbor-joining clustering, based on Nei's genetic distance (Fig 1). The 292 accessions were broadly categorized into two groups: Group I and Group II. Notably, the tea accessions from the test regions appeared in both groups and were not classified according to geographical distance. This suggests that the genetic diversity among the accessions is quite similar, likely due to their close geographical proximity, which has resulted in the clustering of these accessions.

A total of 292 tea plant accessions were analyzed using principal coordinate analysis (PCoA) with GenAlEx version 6.503 to validate the findings of the phylogenetic examination. The PCoA indicated that the first two coordinates explained 5.66% and 5.48% of the total genetic variation, respectively (Fig 2). This analysis demonstrated that the tea accessions were grouped into two distinct clusters, corroborating the conclusions drawn from the phylogenetic study.

The analysis of the population's genetic structure revealed the highest ΔK value at K=2, suggesting that the model's fit was most accurate when the accessions were categorized into two groups (Fig 3).

The accessions in Cluster I and Cluster II comprise diverse populations that closely align with the results obtained from Neighbor-Joining cluster. These findings suggest that there is minimal variation in the genetic diversity of tea accessions in Huangshan city.

## Core collection development and assessment

Nine candidate core collections of varying sizes were generated using the Core Hunter V3.0 software. To identify the most efficient core size, these nine core sizes were compared against the entire population based on six genetic diversity parameters: Na, Ne, Ho, He, uHe, and I (Table 4).

**Table 3. Genetic diversity parameters for six populations.**

| POP | N | Na | Ne | I | Ho | He | uHe | F |
|---|---|---|---|---|---|---|---|---|
| HS | 50.00 | 12.60 | 6.67 | 2.09 | 0.39 | 0.83 | 0.84 | 0.54 |
| HZ | 42.00 | 11.80 | 6.37 | 2.07 | 0.29 | 0.83 | 0.84 | 0.65 |
| QM | 49.00 | 12.30 | 6.38 | 2.07 | 0.30 | 0.83 | 0.84 | 0.64 |
| SX | 69.00 | 13.60 | 6.51 | 2.12 | 0.37 | 0.83 | 0.84 | 0.55 |
| XN | 48.00 | 12.30 | 6.32 | 2.04 | 0.31 | 0.82 | 0.83 | 0.62 |
| YX | 34.00 | 11.60 | 6.73 | 2.09 | 0.32 | 0.84 | 0.85 | 0.62 |

HS: Huangshan district, HZ: Huizhou district, QM: Qimen county, SX: She county, XN: Xiuning county, YX: Yi county, N: number of individuals, Na: number of alleles, Ne: number of effective alleles, Ho: observed heterozygosity, He: expected heterozygosity, uHe: Unbiased Expected Heterozygosity = [2N/(2N − 1)] × He, I: Shannon's information index, F: inbreeding coefficient = (He − Ho)/He = 1 − (Ho/He).

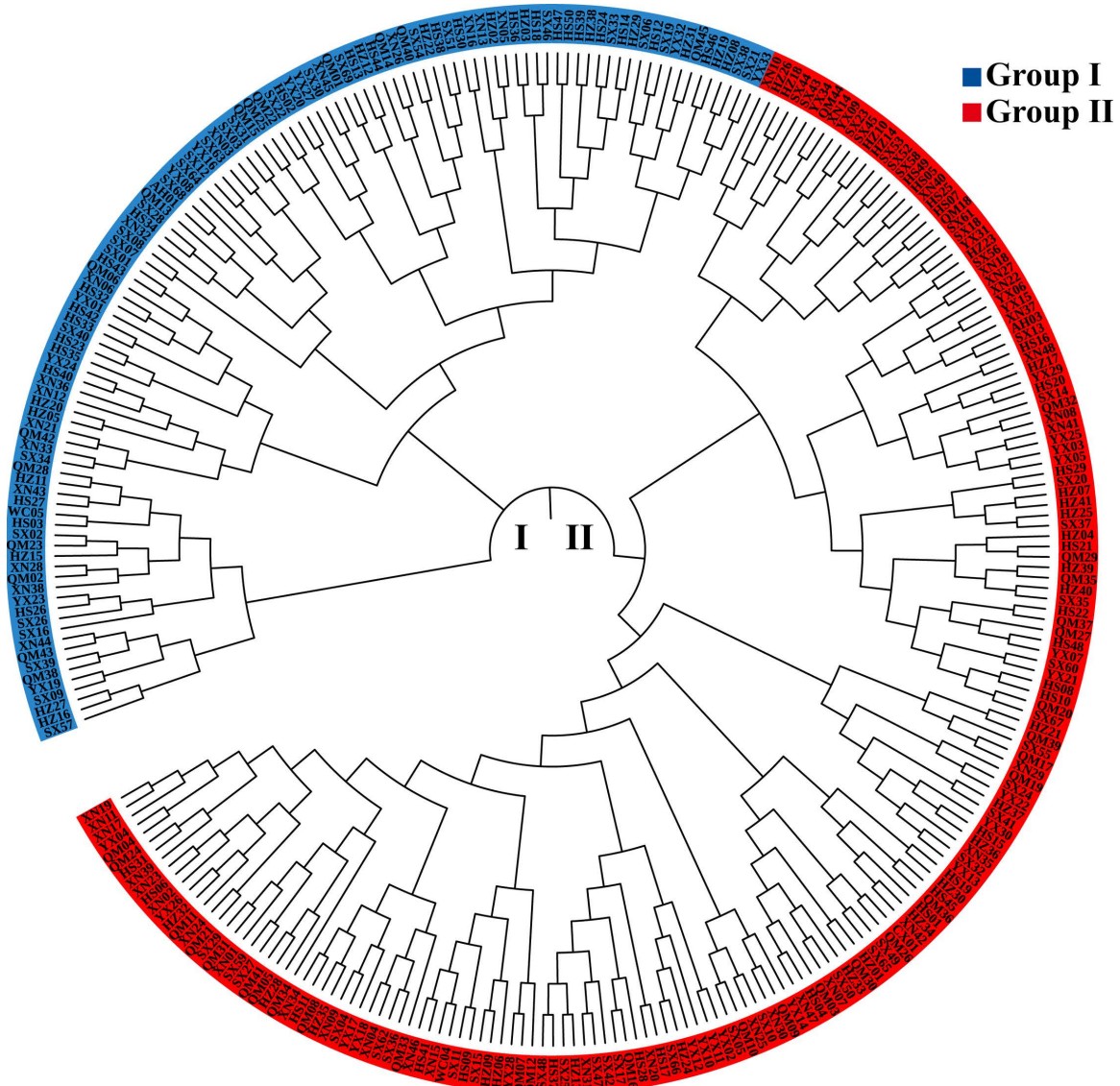

**Fig 1. Neighbor-Joining cluster diagram of 292 tea accessions.** The Neighbor-Joining tree was constructed based on Nei's genetic distance. Group I (blue) includes 112 accessions, and Group II (red) includes 180 accessions.

The core sizes comprising 10% of the total germplasm were found to be significantly different (P ≤ 0.05) from the full population regarding the effective population size (Ne) and expected heterozygosity (He). Additionally, core sizes representing 15% of the total exhibited significant differences (P ≤ 0.05) in the fragrance information index (I). Specifically, when core collections constituted 10% of the total germplasm, the genetic diversity parameters Ne and He demonstrated significant differences (P ≤ 0.05). Conversely, at 15% core collection size, the fragrance information index I displayed significant differences. Furthermore, when the number of alleles accounted for 35%, the number of alleles (Na) showed significant differences. However, other parameters, including observed heterozygosity (Ho), unbiased expected heterozygosity (uHe), and fixation index (F), did not exhibit substantial differences at any percentage.

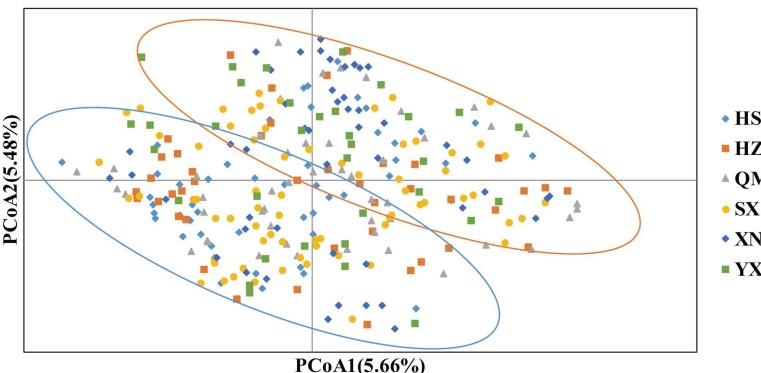

**Fig 2. Principal coordinate analysis of 292 tea accessions.** Due to the close relationship, all the accessions were not distinguished, and could be divided into two groups by auxiliary lines.

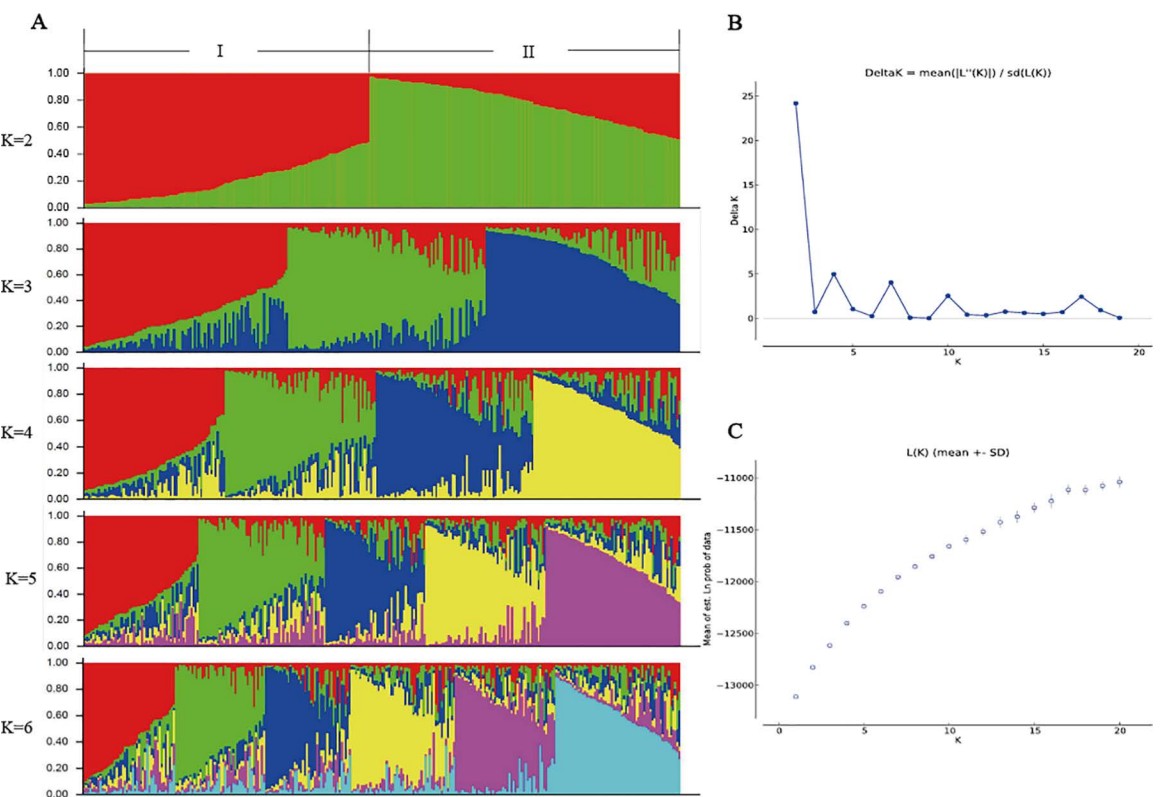

**Fig 3. Structure analysis of 292 tea accessions. (A)** The population structure of tea plants was determined using STRUCTURE 2.3.4 software (K = 2–6), **(B)** Estimated average likelihood L(K) distribution (mean ± SD) from 2 to 20 possible clusters **(K)**, **(C)** Delta K distribution based on the rate of change in L(K) between continuous K values.

As a result, it was determined that a core size of 35% represented the most optimal core germplasm collection, comprising 102 tea accessions. This core collection included 17 accessions from QM, 22 from SX, 14 from XN, 14 from YX, 18 from HS, and 15 from HZ (Fig 4B). The composition was as follows (S2 Table). In comparison to the entire collection,

**Table 4. Comparison of genetic diversity parameters of different fractions of core collections.**

| Core collection | Number of accessions | Na | Ne | I | Ho | He | uHe | F |
|---|---|---|---|---|---|---|---|---|
| 100% | 292 | 12.37 | 6.50 | 2.08 | 0.33 | 0.83 | 0.84 | 0.60 |
| 50% | 146 | 11.15 | 7.19 | 2.13 | 0.36 | 0.85 | 0.87 | 0.58 |
| 45% | 131 | 10.98 | 7.26 | 2.13 | 0.37 | 0.85 | 0.87 | 0.57 |
| 40% | 116 | 10.67 | 7.23 | 2.12 | 0.38 | 0.85 | 0.88 | 0.56 |
| 35% | 102 | 10.22 | 7.07 | 2.09 | 0.38 | 0.85 | 0.87 | 0.56 |
| 30% | 87 | 9.62* | 6.85 | 2.04 | 0.38 | 0.84 | 0.87 | 0.55 |
| 25% | 73 | 8.75* | 6.38 | 1.96 | 0.40 | 0.83 | 0.87 | 0.51 |
| 20% | 58 | 8.00* | 6.05 | 1.89 | 0.40 | 0.82 | 0.87 | 0.51 |
| 15% | 43 | 6.87* | 5.56 | 1.77* | 0.40 | 0.80 | 0.87 | 0.51 |
| 10% | 29 | 5.48* | 4.78* | 1.58* | 0.39 | 0.77* | 0.87 | 0.50 |

Na: number of alleles, Ne: number of effective alleles, I: Shannon's information index, Ho: observed heterozygosity, He: expected heterozygosity, uHe: Unbiased Expected Heterozygosity = [2N/(2N − 1)] * He, F: inbreeding coefficient = (He − Ho)/He = 1 − (Ho/He). *$P \leq 0.05$ for the difference between a core subset and the total population of accessions in simple t-tests.

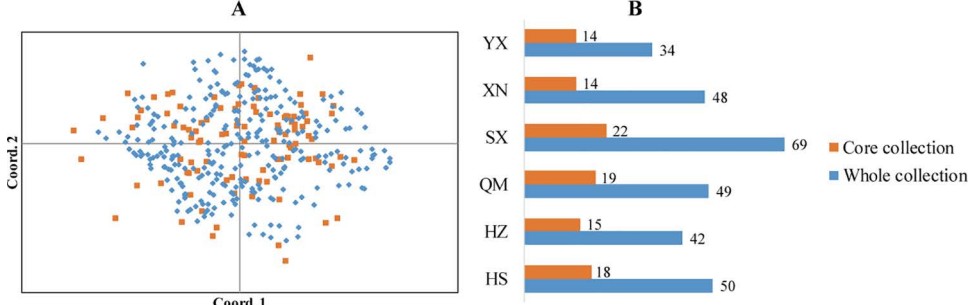

**Fig 4. (A) PCoA of the core collection (n=102) and whole collection (n=292) were based on genotypic data from SSR markers. (B)** Comparison of core collection and initial whole collection.

the core collection maintained 82.62% of Na, 100.48% of I, and 115.15% of Ho. Furthermore, it demonstrated a 108.77% increase in Ne, a 102.41% increase in He, and a 103.57% increase in uHe. The research indicates that the core and the whole collection exhibit a similar distribution (Fig 4A).

## Discussion

### Genetic diversity and population structure

The identification and evaluation of germplasm resources are essential for their protection and utilization, serving as a critical foundation for germplasm innovation, genetic enhancement, and the development of plant breeding [34]. Currently, the population structure and genetic differences of 208 tea landraces in Fujian province have been studied [35]. Additionally, Wang examined the genetic diversity and phylogenetic relationships of 50 ancient tea accessions in Guangxi province [36]. Anhui Province is rich in tea landrace resources, including Huangshan, Qimen, Jinji, and other species [37]. However, there is a scarcity of studies focusing on the population structure and genetic differences of landraces, with most research primarily addressing clonal varieties and related aspects.

In this research, a substantial number of tea landraces from Huangshan city were collected for the first time to analyze their genetic diversity and population structure. The findings indicated that the population from She County exhibited the

highest genetic variability (Na = 13.60, I = 2.12) (Table 4). This is attributed to the fact that She County has the largest tea plantation area in Huangshan city, amounting to 18,900 hectares, with the majority being tea landraces [38]. The slow promotion of clonal tea plants has contributed to the effective protection of the landraces. The genetic distance among the populations in Huangshan city is relatively small, allowing for the population structure to be broadly categorized into two groups. The results of Neighbor-Joining, PCoA, and Structure analyses support this conclusion.

## The core collection of tea accessions

Determining the number of core collections is essential. Currently, the proportion of core collections for woody plants typically ranges from 10% to 45% [39]. For instance, the Chinese fir [31] has a core collection proportion of 42.9% (300 out of 700), while *Toona sinensis* [40] accounts for 20% (208 out of 1040). *Ginkgo biloba L.* [41] has a core collection proportion of 12.3% (27 out of 101), and *Camellia oleifera C.Abel* [42] represents 15% (45 out of 300). In this study, we employed Core Hunter 3.0 software to select 35% of core collections from the accessions of Huangshan city, resulting in a total of 102 accessions. Among these, 19 accessions were selected, representing 17.6% of the QM accessions. Additionally, 22 accessions were selected from SX (21.6%), 14 from XN (13.7%), 14 from YX (13.7%), 18 from HS (17.6%), and 15 from HZ (14.7%).

Allele retention is a crucial factor in evaluating the construction of core germplasm. For instance, when establishing the seed bank for sugarcane (*Saccharum officinarum L.*), the objective was to preserve at least 70% of allele diversity and other indicators of genetic diversity [43]. A comparison between the core collection and the entire collection revealed that the core collection exhibited higher genetic diversity, retaining 82.62% of allele richness. The indicators Ne, I, Ho, He, and uHe not only did not decrease but also showed an increase, indicating a balanced geographical distribution (Table 4). Based on these findings, the construction of core collections offers good representation and coverage. This study represents the first evaluation of genetic diversity in tea landrace resources in Huangshan city, and the successful establishment of core collections provides a foundation for the identification and conservation of tea landrace resources in China.

## Conclusions

In this study, ten pairs of SSR markers were utilized to analyze the genetic diversity and structure of tea accessions in Huangshan city. The results indicated significant genetic diversity among the accessions, with a close genetic distance observed between county populations. This close genetic distance may be attributed to the geographical proximity of the counties. A core collection was established, encompassing 35% of the entire accessions, which included 19 accessions from Qimen County, 22 from She County, 14 from Xiuning County, 14 from Yi County, 18 from Huangshan District, and 15 from Huizhou District. The selection and establishment of this core germplasm provide a crucial foundation and theoretical support for the effective management, protection, and utilization of germplasm resources in Huangshan city.

## Supporting information

**S1 Table. Geographic distribution of 292 collected tea accessions.**
(DOCX)

**S2 Table. Information of the reserved accessions.**
(DOCX)

**S3 Minimal Data Set.**
(ZIP)

## Acknowledgments

The researchers would like to acknowledge the agricultural administration department of Huangshan city to provide the excellent support in this project.

## Author contributions

**Conceptualization:** Xu Ruan, Qiong Wu, Wenjie Wang.

**Data curation:** Xu Ruan, Qiong Wu.

**Funding acquisition:** Wenjie Wang.

**Investigation:** Xu Ruan, Xiaoyu Jiao, Leigang Wang.

**Methodology:** Xu Ruan.

**Project administration:** Qiong Wu, Wenjie Wang.

**Resources:** Wenjie Wang.

**Software:** Dandan Liu.

**Supervision:** Qiong Wu.

**Validation:** Minghui Sun.

**Visualization:** Dandan Liu.

**Writing – original draft:** Xu Ruan.

**Writing – review & editing:** Qiong Wu, Wenjie Wang.

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
