## [Decision Letter · Decision Letter 0]

20 Oct 2024

PONE-D-24-42251Core collection construction and genetic diversity analysis of tea plant (Camellia sinensis [L.] O. Kuntze) accessions in Huangshan city using SSR markersPLOS ONE

Dear Dr. Wang,

Thank you for submitting your manuscript to PLOS ONE. After careful consideration, we feel that it has merit but does not fully meet PLOS ONE’s publication criteria as it currently stands. Therefore, we invite you to submit a revised version of the manuscript that addresses the points raised during the review process.

We look forward to receiving your revised manuscript.

Kind regards,

Mehdi Rahimi, Ph.D.

Academic Editor

PLOS ONE

4. Please include a caption for figure 2, 3, 4.

Additional Editor Comments:

Dear Author

The reviewer(s) have recommended major revisions to your manuscript. Therefore, I invite you to respond to the reviewer(s)' comments and revise your manuscript.

With Thanks

Reviewers' comments:

Reviewer's Responses to Questions

**Comments to the Author**

1. Is the manuscript technically sound, and do the data support the conclusions?

Reviewer #1: Yes

Reviewer #2: Yes

2. Has the statistical analysis been performed appropriately and rigorously? 

Reviewer #1: Yes

Reviewer #2: Yes

3. Have the authors made all data underlying the findings in their manuscript fully available?

Reviewer #1: No

Reviewer #2: Yes

4. Is the manuscript presented in an intelligible fashion and written in standard English?

Reviewer #1: Yes

Reviewer #2: Yes

5. Review Comments to the Author

Reviewer #1: It is positive to work on such large number accession. However, the number of markers used in the study is low. Moreover, we have currently in the era of NGS that can yield thousands of SNP markers. SSR markers are mainly found in areas of non coding region. Thus, have less significance in comparison to SNP markers. The main purpose of constructing core collection is to effectively address the conflict between a large number of germplasm resources in gene banks and their effective preservation, to improve the utilization rate of excellent germplasm, to promote the exploration of excellent gene and molecular markers. Core collections have to contain the largest portion of variation for certain key traits so that they would be utilizes by the breeder. Making core collection using SSR marker have shortcomings in capturing the variations/diversity in key phenotypic traits (resulted from natural or artificial selection). I strongly suggest this work to be supported by phenotypic (field) data or use more robust marker (SNP).

Reviewer #2: I have thoroughly read the article. This study presents a well-structured and comprehensive analysis of tea genetic diversity, particularly in the Anhui region. The introduction effectively highlights the challenges associated with the decline of local landraces and the importance of conservation efforts. The materials and methods section is detailed and thorough, providing enough information for replication, which enhances the study's reliability. The results offer valuable insights, particularly the high genetic diversity in She County and the division of population structure into two distinct groups, supported by multiple analytical approaches. The creation of a core collection retaining 82.62% of allele richness is commendable, though further explanation on core collection selection would strengthen the findings.

6. PLOS authors have the option to publish the peer review history of their article (what does this mean? ). If published, this will include your full peer review and any attached files.

**Do you want your identity to be public for this peer review?** For information about this choice, including consent withdrawal, please see our Privacy Policy .

Reviewer #1: No

Reviewer #2: No

---

## [Author Response · Author response to Decision Letter 0]

29 Oct 2024

Editor Comments:

Response: We have thoroughly rechecked the manuscript and corrected it as per the PLOS template that is provided.

Response: The plant materials for the experiment were obtained with the help of the agricultural authorities of the counties and districts of Huangshan city. We communicated sufficiently with them in the early stages, so no relevant permit was required.

3. We note that Figure 1 in your submission contain [map/satellite] images which may be copyrighted. All PLOS content is published under the Creative Commons Attribution License (CC BY 4.0), which means that the manuscript, images, and Supporting Information files will be freely available online, and any third party is permitted to access, download, copy, distribute, and use these materials in any way, even commercially, with proper attribution. For these reasons, we cannot publish previously copyrighted maps or satellite images created using proprietary data, such as Google software (Google Maps, Street View, and Earth). For more information, see our copyright guidelines: http://journals.plos.org/plosone/s/licenses-and-copyright. We require you to either (1) present written permission from the copyright holder to publish these figures specifically under the CC BY 4.0 license, or (2) remove the figures from your submission:1. You may seek permission from the original copyright holder of Figure 1 to publish the content specifically under the CC BY 4.0 license. We recommend that you contact the original copyright holder with the Content Permission Form (http://journals.plos.org/plosone/s/file?id=7c09/content-permission-form.pdf) and the following text: “I request permission for the open-access journal PLOS ONE to publish XXX under the Creative Commons Attribution License (CCAL) CC BY 4.0 (http://creativecommons.org/licenses/by/4.0/). Please be aware that this license allows unrestricted use and distribution, even commercially, by third parties. Please reply and provide explicit written permission to publish XXX under a CC BY license and complete the attached form.” Please upload the completed Content Permission Form or other proof of granted permissions as an ""Other"" file with your submission. In the figure caption of the copyrighted figure, please include the following text: “Reprinted from [ref] under a CC BY license, with permission from [name of publisher], original copyright [original copyright year].” 2. If you are unable to obtain permission from the original copyright holder to publish these figures under the CC BY 4.0 license or if the copyright holder’s requirements are incompatible with the CC BY 4.0 license, please either i) remove the figure or ii) supply a replacement figure that complies with the CC BY 4.0 license. Please check copyright information on all replacement figures and update the figure caption with source information. If applicable, please specify in the figure caption text when a figure is similar but not identical to the original image and is therefore for illustrative purposes only. The following resources for replacing copyrighted map figures may be helpful: USGS National Map Viewer (public domain): http://viewer.nationalmap.gov/viewer/, The Gateway to Astronaut Photography of Earth (public domain): http://eol.jsc.nasa.gov/sseop/clickmap/, Maps at the CIA (public domain): https://www.cia.gov/library/publications/the-world-factbook/index.html and https://www.cia.gov/library/publications/cia-maps-publications/index.html, NASA Earth Observatory (public domain): http://earthobservatory.nasa.gov/, Landsat: http://landsat.visibleearth.nasa.gov/, USGS EROS (Earth Resources Observatory and Science (EROS) Center) (public domain): http://eros.usgs.gov/#, Natural Earth (public domain): http://www.naturalearthdata.com/.

Response: As we have recently learned, inserting a map into a paper is a very complex task, including surveying department review, copyright licensing, and so on. Therefore, we removed Figure 1 and replaced it with S1 Table to illustrate the accessions sampling information, and the subsequent figure number changed accordingly.

4. Please include a caption for figure 2, 3, 4.

Response: We have included captions for figure 2, 3, and 4 (as figure 1 has been removed, the picture numbers have been changed to figure 1, 2, 3), and we also revised the table number. In addition, we have improved figure 2 to match figure 4 (Line 177).

Reviewer comments

Reviewer #1: It is positive to work on such large number accession. However, the number of markers used in the study is low. Moreover, we have currently in the era of NGS that can yield thousands of SNP markers. SSR markers are mainly found in areas of non coding region. Thus, have less significance in comparison to SNP markers. The main purpose of constructing core collection is to effectively address the conflict between a large number of germplasm resources in gene banks and their effective preservation, to improve the utilization rate of excellent germplasm, to promote the exploration of excellent gene and molecular markers. Core collections have to contain the largest portion of variation for certain key traits so that they would be utilizes by the breeder. Making core collection using SSR marker have shortcomings in capturing the variations/diversity in key phenotypic traits (resulted from natural or artificial selection). I strongly suggest this work to be supported by phenotypic (field) data or use more robust marker (SNP).

Response: Thank you so much for the comments.

1. In this experiment, capillary gel electrophoresis was used. The number of samples was large, and the experiment cost was high. Therefore, considering the reasons for saving money, we selected 10 pairs of primers. However, the 10 pairs of primers were selected layer by layer from 50 pairs of primers to ensure that they could amplify and reflect the genetic diversity of the sample.

2. At present, SNP molecular marker technology has many advantages, such as a high density of distribution, a high degree of association with functional genes, and strong genetic stability. However, this experiment is carried out based on the last SSR marker experiment, and has some relevance, so we continue to select SSR markers for the experiment. Thank you very much for your strong suggestion. Next, we will try to use SNP markers combined with field phenotypic data to conduct a study on Anhui Landraces Germplasm Tea Repository, expecting more influential results.

Reviewer #2: I have thoroughly read the article. This study presents a well-structured and comprehensive analysis of tea genetic diversity, particularly in the Anhui region. The introduction effectively highlights the challenges associated with the decline of local landraces and the importance of conservation efforts. The materials and methods section is detailed and thorough, providing enough information for replication, which enhances the study's reliability. The results offer valuable insights, particularly the high genetic diversity in She County and the division of population structure into two distinct groups, supported by multiple analytical approaches. The creation of a core collection retaining 82.62% of allele richness is commendable, though further explanation on core collection selection would strengthen the findings.

Response: Thank you so much for the comments. We are grateful for the reviewer’s kind comments, and your encouragement will make us do better in the future.

---

## [Editor Report · Decision Letter 1]

18 Mar 2025

Core collection construction and genetic diversity analysis of tea plant (Camellia sinensis [L.] O. Kuntze) accessions in Huangshan city using SSR markers

PONE-D-24-42251R1

Dear Dr. Wang,

We’re pleased to inform you that your manuscript has been judged scientifically suitable for publication and will be formally accepted for publication once it meets all outstanding technical requirements.

Kind regards,

Mehdi Rahimi, Ph.D.

Academic Editor

PLOS ONE
---

## [Editor Report · Acceptance letter]

PONE-D-24-42251R1

PLOS ONE

Dear Dr. Wang,

I'm pleased to inform you that your manuscript has been deemed suitable for publication in PLOS ONE. Congratulations! Your manuscript is now being handed over to our production team.

Kind regards,

on behalf of

Associate Prof. Mehdi Rahimi

Academic Editor

PLOS ONE